



# Stepwise chemical abrasion ID-TIMS-TEA of microfractured Hadean zircon

C. Brenhin Keller[1], Patrick Boehnke[2], Blair Schoene[3], and T. Mark Harrison[4]

[1]Department of Earth Sciences, Dartmouth College, Hanover, NH 03755
[2]Eta Vision, Chicago, IL 60611
[3]Department of Geosciences, Guyot Hall, Princeton University, Princeton, NJ 08544
[4]Department of Earth, Planetary, and Space Sciences, University of California, Los Angeles, CA 90095

**Correspondence:** C. Brenhin Keller (cbkeller@dartmouth.edu)

**Abstract.**

The Hadean Jack Hills zircons represent the oldest known terrestrial material, providing a unique and truly direct record of Hadean Earth history. This zircon population has been extensively studied via high spatial resolution, high throughput *in situ* isotopic and elemental analysis techniques such as secondary ionization mass spectrometry (SIMS), but not by comparatively

destructive, high-temporal-precision (<0.05% two-sigma) thermal ionization mass spectrometry (TIMS). In order to better understand the lead loss and alteration history of terrestrial Hadean zircons, we conduct stepwise chemical abrasion isotope dilution thermal ionization mass spectrometry with trace element analysis (CA-ID-TIMS-TEA) on manually microfractured Hadean Jack Hills zircon fragments previously dated by SIMS. We conducted three successive HF leaching steps on each individual zircon fragment, followed by column chromatography to isolate U-Pb and trace element fractions. Following iso-

topic and elemental analysis, the result is an independent age and trace element composition for each leachate of each zircon fragment. We observe ∼50 Myr of age heterogeneity in concordant residues from a single zircon grain, along with a protracted history of post-Hadean Pb-loss with at least two modes circa ∼0 and 2-4 Ga. Meanwhile, step-wise leachate trace element chemistry reveals enrichments of light rare earth elements, uranium, thorium, and radiogenic lead in early leached domains relative to the zircon residue. In addition to confirming the efficacy of the *LREE-I* alteration index and providing new insight

into the mechanism of chemical abrasion, the interpretation and reconciliation of these results suggests that Pb-loss is largely driven by low-temperature aqueous recrystallization, and that regional thermal events may act to halt – not initiate – Pb-loss from metamict domains in the Hadean Jack Hills zircons.

## 1 Introduction

Terrestrial zircons with U-Pb ages in excess of 4 Ga were first fortuitously discovered in the Paleoarchean Mt. Narryer quartzite

by Froude et al. (1983), and subsequently in greater abundance by Compston and Pidgeon (1986) in a quartz pebble metaconglomerate at the Jack Hills – both in the Narryer Gneiss Complex of the Yilgarn Craton, western Australia. Zircons with Hadean (> 4 Ga) $^{207}$Pb/$^{206}$Pb ages have subsequently been reported from most other continents including North America (Bowring and Williams, 1999; Mojzsis and Harrison, 2002; Iizuka et al., 2006), South America (Nadeau et al., 2013; Paquette





et al., 2015), Eurasia (Wang et al., 2007; Duo et al., 2007; Xu et al., 2012; Xing et al., 2014), India (Miller et al., 2018), and Africa (Byerly et al., 2018), suggesting a widely distributed occurrence of zircon-bearing crust by at least the late Hadean. Nonetheless, both the antiquity (Valley et al., 2014) and quantity (Holden et al., 2009) of Hadean zircon from the Jack Hills far exceeds that yet analyzed from any other locality; as such, the Jack Hills zircon record predominates our understanding of the
Hadean Eon on Earth.

While the interpretation of petrologic and geochemical data derived from Hadean zircons can be difficult, many constraints have been interpreted to suggest a relatively temperate Hadean eon, featuring liquid water and continental crust (Cavosie et al., 2007; Harrison, 2009; Harrison et al., 2017). Hadean Jack Hills zircons display oxygen isotope compositions enriched in $^{18}$O relative to the mantle, suggesting a parental magma that incorporated silicates which have interacted with liquid water
(Mojzsis et al., 2001; Wilde et al., 2001; Cavosie et al., 2005; Trail et al., 2007). Unlike lunar and meteoritic zircon (Hoskin, 2003), Jack Hills Hadean zircons display positive Ce anomalies (Trail et al., 2011; Bell et al., 2016), suggesting conditions sufficiently oxidized to produce $Ce^{4+}$, perhaps associated with magmatic water. Although magma Ti activity is not perfectly constrained for detrital zircons (except in a handful of zircons containing apparently primary rutile inclusions), observed Ti-in-zircon temperatures of ~680 C are most consistent with a parental magma produced by water-saturated eutectic melting
of pelitic sediment (Watson, 2005; Harrison, 2009). The same Hadean zircons display felsic inclusion suites including some phases such as apatite, biotite, hornblende, and alkali feldspar (Maas et al., 1992; Hopkins et al., 2010; Bell et al., 2015) that are not abundant or not reported in the host quartzite (Myers, 1988) but are ubiquitous components of granitic magmas. Compounding the above constraints, higher mantle potential temperatures in the Hadean imply lower zirconium abundances for a given magma $SiO_2$, increasing the difficulty of saturating zircon and increasing the volume of felsic crust required to
crystallize a given volume of zircon (Keller et al., 2017). If correct, such a relatively uniformitarian Hadean would appear plausibly consistent with independent evidence for subduction-driven flux melting since at least 3.85 Ga (Keller and Schoene, 2018). Nonetheless, a large proportion of the Archean geological community would strongly dispute such views (Condie, 2018; Bédard, 2018), and controversy regarding the nature and origin of Earth's earliest crust is likely to persist. Consequently, much remains to be learned from the terrestrial Hadean zircon record.

To date, the study of the Jack Hills zircons has proceeded in tandem with the development of high throughput, minimally destructive *in situ* analytical techniques such as Secondary Ion Mass Spectrometry (SIMS) (Froude et al., 1983; Compston and Pidgeon, 1986; Holden et al., 2009). While the high spatial precision and high throughput of these techniques has been critical to the study of the Jack Hills zircons, technical (matrix effects, mass and elemental fractionation) and mathematical (counting statistics) constraints frequently impose an effective tradeoff between spatial and temporal precision.

Consequently, while Hadean $^{207}Pb/^{206}Pb$ ages are frequently resolved to the $\pm 0.5\%$ level, there is a limit to the extent to which the concordance of the independent $^{206}Pb/^{238}U$ and $^{207}Pb/^{235}U$ chronometers (and thus our confidence that a measured age reflects closed-system behavior) can be established with *in situ* methods. Such limitations are particularly relevant when attempting to distinguish early open-system behavior (i.e., early Pb-loss or Pb-gain), which will move samples nearly parallel to Concordia (Wetherill, 1956). Early Pb-remobilization during Archean ultra-high-temperature (UHT) metamorphism has
been observed in at least one case to produce spurious apparent Hadean $^{207}Pb/^{206}Pb$ ages in Eoarchean zircons from UHT



granulites of the Napier Complex, Enderby Land, Antarctica (Kusiak et al., 2013; Kelly and Harley, 2005). However, such extreme effects have been ruled out in the Jack Hills zircons (Valley et al., 2014) which do not appear to have undergone greater than greenschist facies metamorphism (Trail et al., 2016). Even so, early Pb mobility – particularly Pb loss – has often been considered as a limitation when interpreting Hadean zircon Hf isotope systematics (Guitreau and Blichert-Toft, 2014; Bell et al., 2014; Whitehouse et al., 2017)

While originally requiring multi-grain zircon aliquots some eight orders of magnitude larger than a typical SIMS ablation pit, the average mass of sample used in a bulk isotope dilution TIMS U-Pb analysis has decreased by more than five orders of magnitude between 1975 and 2010. Over the the same period, temporal precision has improved by over an order of magnitude, all due to improvements in analytical techniques and instrumentation (Schoene, 2014). In total, we may now expect to obtain $< 0.05$ % relative temporal precision and accuracy on a single $< 1$ $\mu g$ fragment of Hadean zircon, providing a precise and accurate test of closed-system behavior through concordance.

To improve the likelihood of analyzing closed-system material, zircon fragments intended for ID-TIMS may be first treated with chemical abrasion, which has been observed to selectively dissolve damaged domains likely to have undergone Pb-loss (Mattinson, 2005; Mundil et al., 2004). While twelve hours of chemical abrasion in concentrated HF at 210 °C is frequently presumed to effectively mitigate Pb-loss in zircon, the underlying mechanism and the kinetics of this process remain poorly understood. Moreover, since previously published TIMS ages for Jack Hills Hadean zircons (Amelin et al., 1999) predate the advent of chemical abrasion, it was unknown whether such Hadean zircons could survive the full standard 12 hr / 210 °C chemical abrasion procedure. Conducting chemical abrasion in a stepwise manner, where intermediate leachates are extracted and retained for analysis, eliminates this risk. By combining such *stepwise* chemical abrasion with TIMS-TEA, we may obtain matched trace-element and geochronological data for each subsequent chemical abrasion step of each analyzed zircon fragment. While time-consuming, such an analytical procedure has the potential to provide insight into both the geologic history of Jack Hills Hadean zircon and the efficacy of chemical abrasion.

## 2 Methods

Here we apply stepwise CA-ID-TIMS-TEA (chemical abrasion, isotope dilution, thermal ionization mass spectrometry with trace element analysis) to sub-grain fragments of Jack Hills zircons. Since only some three percent of Jack Hills zircons have ages >4.0 Ga (Harrison, 2009), Jack Hills zircons with late Hadean ($\sim 4.0 - 4.1$ Ga) SIMS ages were selected from epoxy mounts previously characterized by *in situ* techniques at UCLA (Table S1). A total of 23 epoxy mounted half-zircons were selected for TIMS analysis at Princeton University, of which 14 were further dissected into two to five fragments each by microfracturing with a tungsten carbide point, resulting in a grand total of 54 sub-grain zircon fragments.

To prepare for chemical abrasion (Mattinson, 2005), each zircon fragment was individually loaded into a separate quartz crucible and annealed for 48 hours at 900 °C. Annealed zircons were transferred to 3 ml Savillex perfluoroacetate (PFA) beakers and moved to a class 1000 cleanroom where they were rinsed with MilliQ ultrapure water, transferred to 200 $\mu l$ Savillex PFA microcapsules, and rinsed with ultrapure HCl. Subsequent analytical steps were conducted in the cleanroom using class





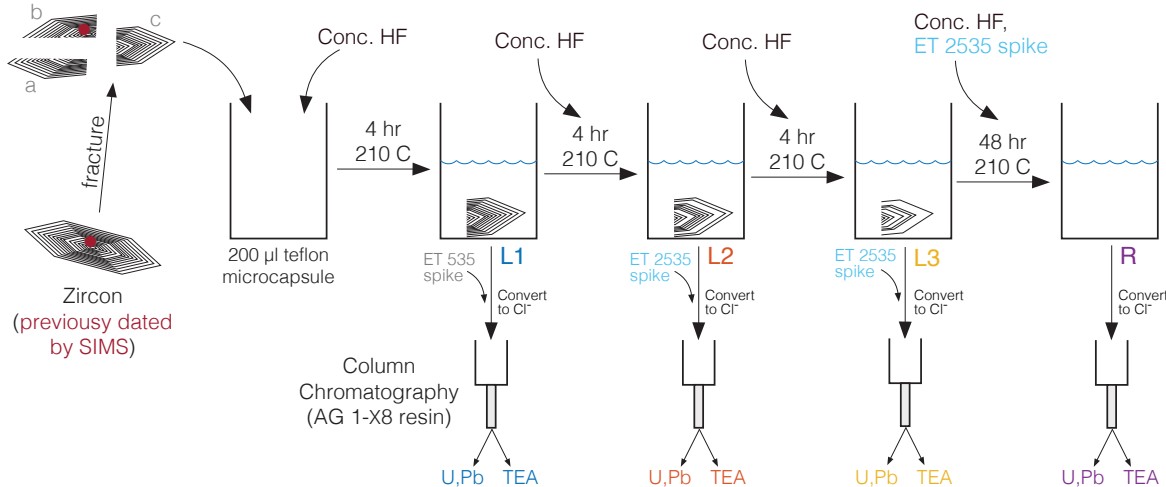

**Figure 1.** Schematic illustration of the step leaching methodology employed in this study. U, Pb fractions separated by column chromatography for each leachate of each zircon fragment were analyzed on an IsotopX Phoenix 62 Thermal Ionization Mass Spectrometer, while TEA solutions were analyzed for trace element concentration on a Thermo Element 2 ICPMS at Princeton University.

10 clean hoods, ultrapure reagents distilled in a Savillex DST-1000 sub-boiling still (blank-checked to ensure common Pb concentrations less than 0.1 pg/g), and PFA labware cleaned by heating with alternating ultrapure acids for periods of months to years.

In the first analytical campaign, 36 zircon fragments in separate microcapsules were loaded into two teflon-lined Parr pres-
5   sure dissolution vessels with ultrapure hydrofluoric acid (100 $\mu l$ 29 M HF plus 15 $\mu l$ 3 M $HNO_3$ per microcapsule, with 5 ml moat HF) and chemically abraded in two steps of six hours at 210 °C. In the second analytical campaign, the remaining 18 zircon fragments were chemically abraded in a single Parr vessel in three steps of four hours, as illustrated in Figure 1. Between each leaching step, all supernatant acid was extracted, spiked, and retained for analysis (comprising the L1, L2, and L3 leachates). Subsequently, the surviving zircon residue was thoroughly rinsed with $H_2O$, HCl, $HNO_3$, and HF, before finally
10   dissolving any surviving zircon over 48 hours at 210 °C with ultrapure HF (as during abrasion) and a measured quantity of iso-tope dilution tracer. The EARTHTIME $^{205}Pb$-$^{233}U$-$^{235}U$ "ET535" tracer (Condon et al., 2015; McLean et al., 2015) was used for all L1 analyses, while the EARTHTIME $^{202}Pb$-$^{205}Pb$-$^{233}U$-$^{235}U$ double-spike "ET2535" was used for the more critical L2, L3, and residue analyses.

After chemical abrasion and dissolution, each of the resulting 54 dissolved zircon residues and 126 leachates was evaporated
15   to dryness, converted to chlorides by heating with ultrapure 6 M HCl, evaporated a second time, and redissolved in ultrapure 3 M HCl to prepare for ion chromatography. For each sample, a small PTFE column was loaded with 50 $\mu l$ of chloride form Eichrom AG1-X8 anion exchange resin (200-400 mesh), cleaned alternately with $H_2O$ and 6 M HCl, and conditioned with 3.1 M HCl. Following the separation procedure of Krogh (1973) with the modifications of Schoene et al. (2010), samples were





loaded and trace elements eluted in 3 M HCl followed by Pb elution in 6 M HCl and U elution in $H_2O$. Eluted U-Pb separates were evaporated to dryness with $\sim$2 $\mu l$ 0.03 M $H_3PO_4$ and stored for analysis.

Isotopic and trace element analyses of the resulting separates were conducted in 2015-2016 at Princeton University. Evaporated U-Pb separates were loaded (U and Pb together) onto zone-refined rhenium filaments with $\sim$2 $\mu l$ silica gel emitter (Gerstenberger and Haase, 1997) for analysis by IsotopX Phoenix 62 TIMS. Thermal ionization mass spectrometry and data reduction procedures were equivalent to those of Schoene et al. (2015), with Pb collected by peak-hopping on a Daly detector, correcting for a detector deadtime of 43.5 ns as determined by repeated analyses of NBS 982 reference material. Where beam intensity allowed, U was collected by static multicollection on Faraday cups with $10^{12}$ $\Omega$ amplifiers; otherwise, U was collected by peak-hopping on a Daly detector, correcting for 37.5 ns deadtime as established by repeated analyses of CRM U500. During TIMS analysis, two fragments were identified as contamination introduced during single-fragment annealing, and rejected. Isotopic data was processed and analytical uncertainty propagated using Tripoli and U-Pb Redux (McLean et al., 2011; Bowring and McLean, 2011), using a $^{238}U/^{235}U$ ratio of 137.818 $\pm$ 0.045 (two-sigma) (Hiess et al., 2012). Trace element separates were subsequently analyzed on a Thermo Scientific Element 2 ICPMS following the procedure of Schoene et al. (2010), with zircon trace element abundances normalized to 496000 ppm Zr in zircon. Finally, zircon U and Th concentrations were calculated using the zircon Th/U ratio determined from Pb isotopic composition, the ICPMS-derived Th concentrations, and ID-TIMS U and Pb masses. The resulting elemental and isotopic data are tabulated in Tables S1 and S2; all analytical uncertainties are reported as two-sigma unless otherwise noted.

## 3  Results

The Concordia diagrams of Figure 2 reveal a highly heterogeneous age population, including four concordant Hadean residues with $^{207}Pb/^{206}Pb$ dates ranging from 4142.30 $\pm$ 0.63 to 4004.20 $\pm$ 0.51 Ma (excluding tracer and decay constant uncertainty), a wide range of variably discordant L2-L3 leachates, and a distinct, highly discordant population of L1 leachates. Three of the four concordant Hadean zircon residues are derived from a single grain, RSES58 z6.10, which also yielded three concordant L3 leachates and a single concordant L2 leachate (all Hadean), as highlighted in Figure 2C. These concordant ages from different fragments of a single zircon crystal span some 70 Myr. As may be expected from Mattinson (2005) and the success of CA-TIMS over the subsequent decade, leachates are typically more discordant than residues. L1 leachates in particular are markedly more discordant than other analyses, forming a broad array trending towards a lower intercept at the origin (Figure 2A), as might result from zero-age Pb-loss. Four leachate analyses – all of them L2 leachates – yield negatively discordant ages.

Zircon residues are observed in Figure 3A to be systematically (with only one imprecise exception) older than their respective leachates in $^{207}Pb/^{206}$ space, even at low discordance. For a given zircon fragment, L1-L3 leachates are found to have $^{207}Pb/^{206}Pb$ ages some 10s to 100s of Myr younger than residues, with the age gap between corresponding leachates and residues increasing with leachate discordance. In particular, since modern U or Pb remobilization (e.g., Pb-loss without



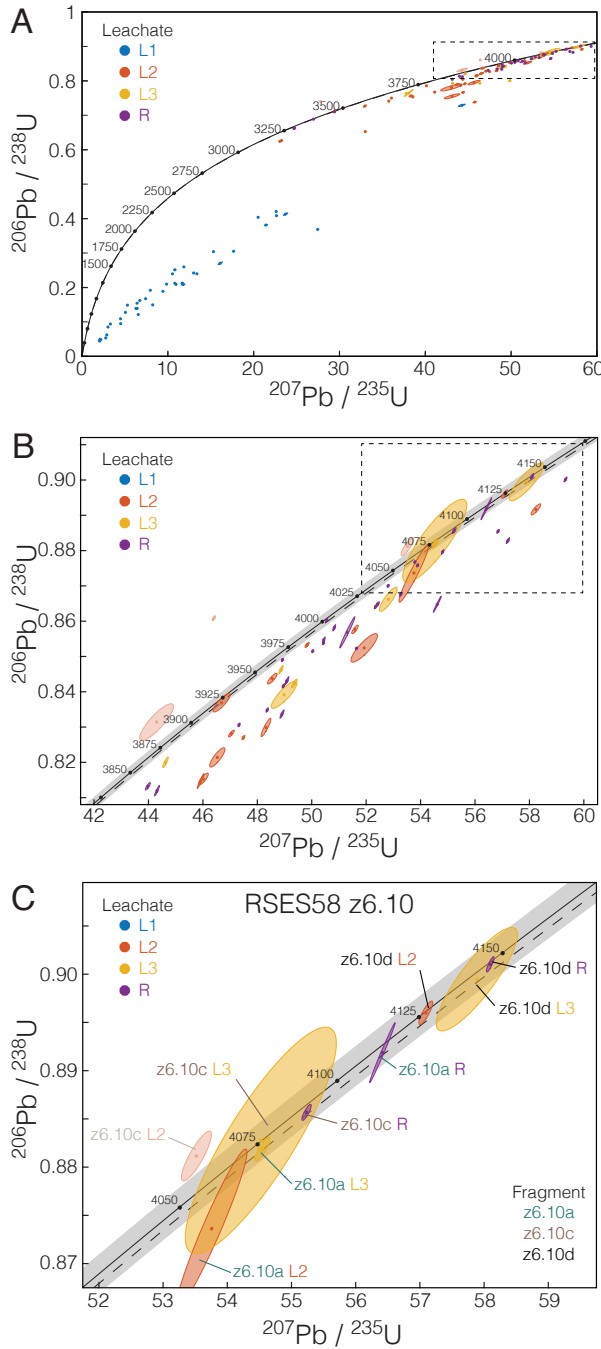

**Figure 2.** CA-ID-TIMS ages for Jack Hills zircon by fragment and leaching step, in Wetherill (1956) concordia space. A: Full range, including recent Pb-loss array in L1 leachates. B: Hadean-Eoarchean inset, emphasizing complexity of the Hadean record suggesting early lead loss and protracted crystallization history. C: Concordant fragments and leachates of zircon RSES58 z6.10, illustrating ∼50 Myr age heterogeneity between concordant residues of zircon fragments from the same polished half-zircon. At this scale, the uncertainty in the $\lambda_{U-238}/\lambda_{U-235}$ decay constant ratio that defines Concordia becomes important; here the solid Concordia line and grey two-sigma error envelope reflects the values of Jaffey et al. (1971), while the dashed line reflects those of Schoene et al. (2006). All dates plotted along the Concordia line are in Ma.

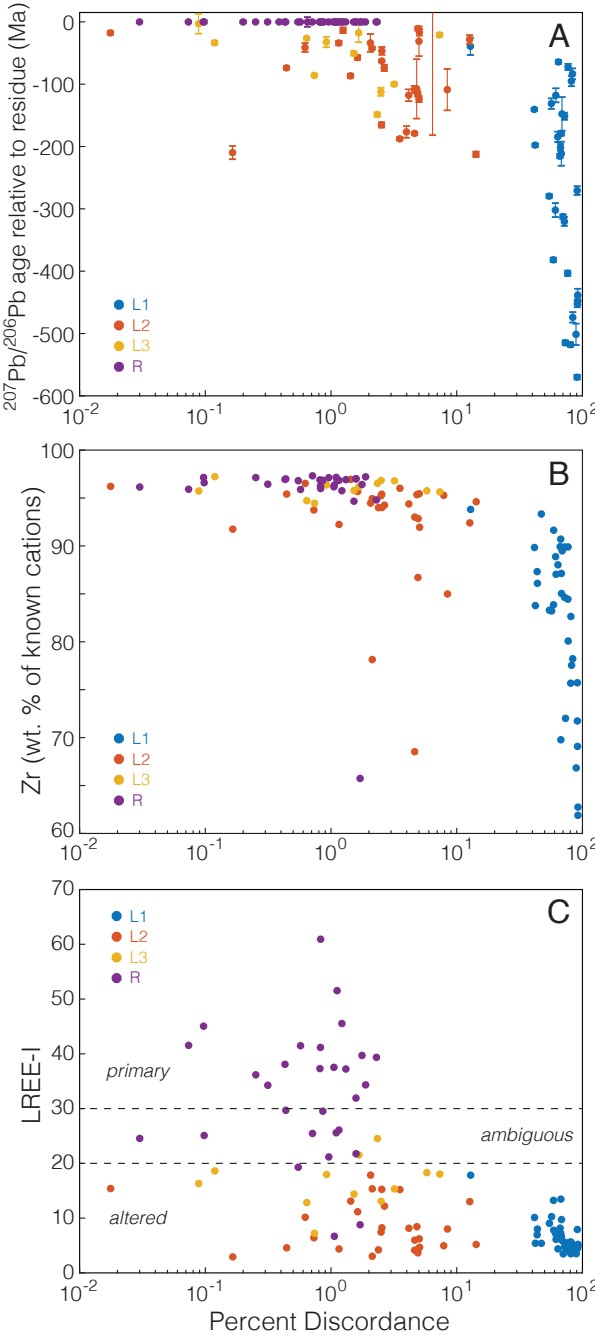

**Figure 3.** A: Age of each leachate relative to its associated residue (if any) plotted as a function of percent discordance. Age offset increases with discordance, but may reach ∼100 Myr even for leachates of similar discordance to their residue. B: Abundance of Zr relative to total measured cation concentration as a function of measured discordance. L1 leachates are distinguished by their high discordance and low Zr proportion. C: The light rare earth alteration index (*LREE-I* = Dy/Nd + Dy/Sm) of Bell et al. (2016, 2019) plotted as a function of measured discordance. High *LREE-I* in TEA measurements accurately distinguishes primary zircon residues from leachates.



additional isotopic fractionation) has no effect on $^{207}Pb/^{206}Pb$ ages, systematic age gaps between residues and leachates in $^{207}Pb/^{206}Pb$ space are most readily attributed to ancient, not recent, Pb-loss.

Using the TIMS-TEA methodology of Schoene et al. (2010), we are able to combine trace element and isotopic analyses on the exact same volume of zircon, allowing us to consider the chemical characteristics of zircons that have undergone

open-system behavior. We observe that both discordance and leaching extent are strongly correlated with bulk chemistry. In particular, L1 leachates are identifiable by their low Zr content as a proportion of measured cations, as well as their extreme discordance. As observed in Figure 3B, Zr represents less than 90% of the observed cation budget by mass in L1 leachate analyses, suggesting that the material removed in L1 leaching steps is not stoichiometric zircon; in later leaching steps, chemistry evolves towards that of the pure zircon residue. Meanwhile, as seen in Figure 3C, leachates are reliably resolved from

pristine residues by the light rare earth index *LREE-I* of Bell et al. (2016, 2019). Reassuringly, all L1 and L2 leachates fall in the "altered" field defined by Bell et al. (2016) (*LREE-I* < 20), while the "primary" (*LREE-I* > 30) field contains only residues; the remaining analyses fall in the "ambiguous" field of *LREE-I* between 20 and 30 comprise residues and L3 leachates.

On an element-by-element basis, we observe a distinct pattern of trace element enrichment in leachates relative to zircon residues (Figure 4). L1 leachates display LREE concentrations up to a factor of 25 higher than their corresponding residues,

along with smaller enrichments in MREE. The discordant L1 leachates are also highly radiogenic, with over ten times the Pb* of pristine zircon residue. Consistent with Pb-loss, this radiogenic lead excess is outpaced by the extreme Th (∼30 x residue) and U (∼50 x residue) concentrations of the same leachates. On the same basis, L2 leachates display comparatively muted enrichments in in REE, U, Th, and Pb*, while L3 leachates display significant enrichments only in LREE.

A comparison of TIMS and SIMS $^{207}Pb/^{206}Pb$ ages in Figure 5 reveals that, for leachates and discordant residues, SIMS

ages (typically targeted on low-U cores) are generally older than TIMS ages on fragments of the same grains. Discordant TIMS analyses, especially including early leachates, are likely accessing damaged open-system domains that were excluded from the analyzed SIMS spot. Indeed, depending on the scale of spatial heterogeneity in U-Pb discordance, smaller analytical volumes may be less likely to mix closed- and open-system domains, leading to increased median concordance as a statistical consequence of smaller analytical volume. However, as seen in Figure 5, TIMS and SIMS $^{207}Pb/^{206}Pb$ ages are in relatively

good agreement for concordant residues surviving the full 12 hours of chemical abrasion.

## 4    Discussion

### 4.1    Chemical Abrasion and U-Pb geochronology

Open-system behavior is arguably the foremost complicating factor in radioisotopic geochronology. With two independent decay chains proceeding at different rates, the U-Pb system in principle allows us to track open-system behavior with discor-

dance, and in some cases to even determine the age of Pb-loss. For zircon, chemical abrasion has been observed to remove damaged domains that have undergone lead loss, and is now widely applied (Mattinson, 2005; Mundil et al., 2004; Mattinson, 2011; Schoene, 2014). However, the same combination of annealing and acid leaching has not been entirely successful in





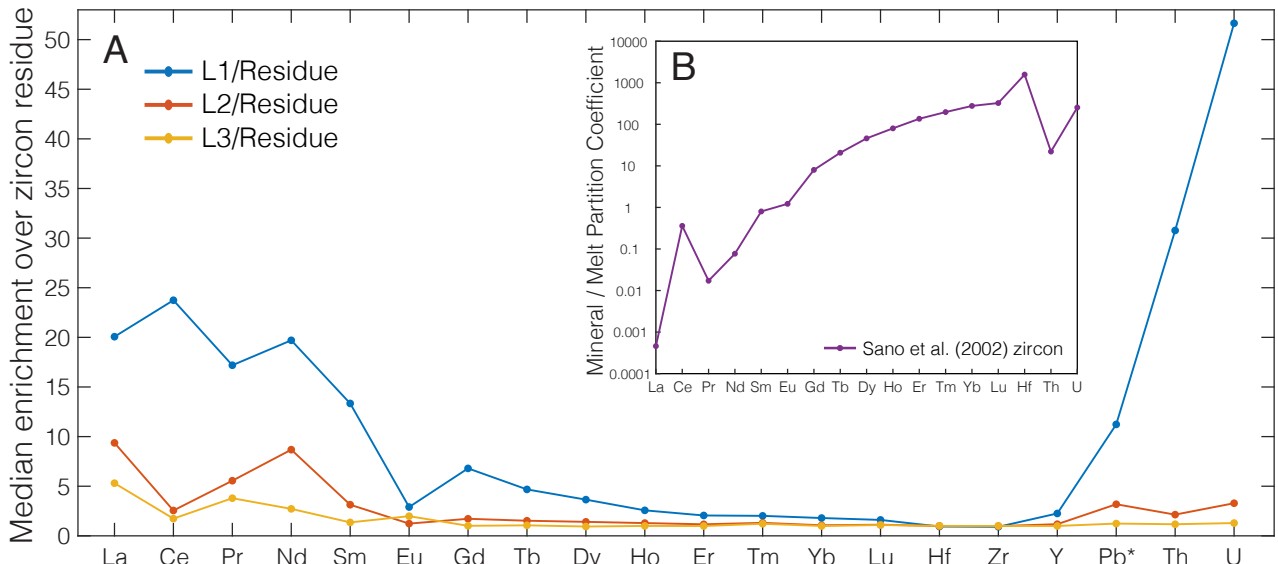

**Figure 4.** A: Chemical abrasion systematically removes zircon domains rich in LREE, Th, U, and radiogenic Pb. This pattern suggests the preferential removal of zones which have undergone radiation damage and metamiction due to high Th, U content, leaving them vulnerable both to geological open-system behaviour (lead loss) and dissolution during chemical abrasion. Negative Eu anomaly and MREE enrichment in L1 leachates may suggest the importance of coupled substitution in the initial formation of high Th, U domains. B: Typical zircon / melt partition coefficients: with the exception of U and Th, the elements that are least abundant in natural zircon display the highest enrichments in L1 leachates

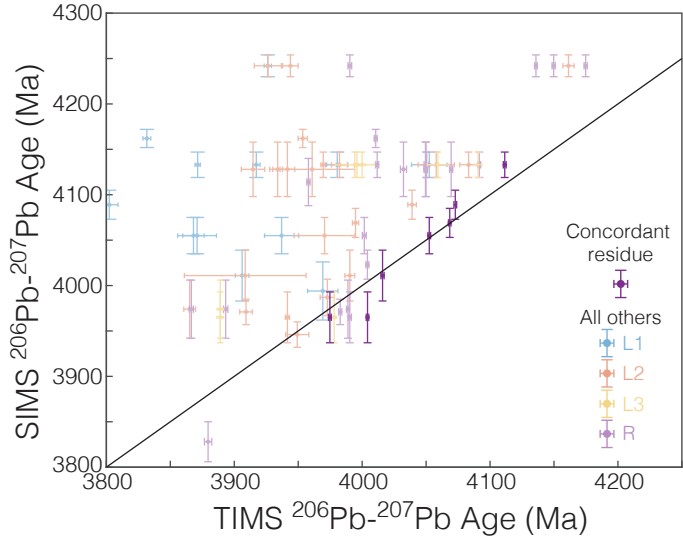

**Figure 5.** Two-variable cross plot of SIMS spot ages and TIMS $^{207}Pb/^{206}Pb$ ages for each fragment and leachate. Concordant residues (bold) plot along the 1:1 line, while others plot above. Horizontal data arrays result from one SIMS spot age per grain plotted against up to four TIMS ages per fragment, with multiple fragments per grain.



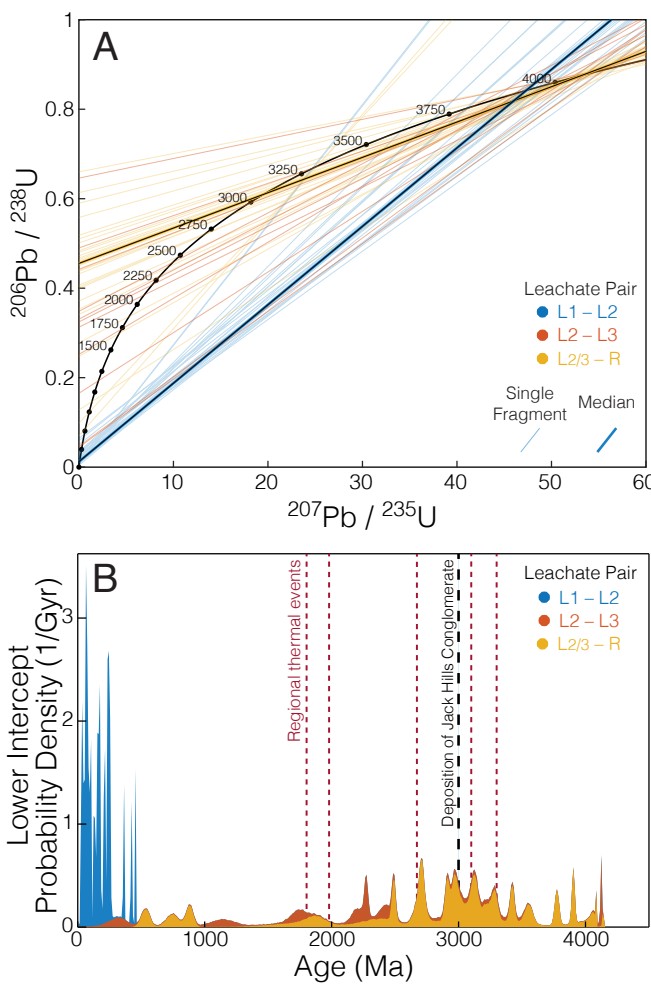

**Figure 6.** A: Discordia arrays defined by pairs of successive leachate and residue analyses from the same zircon fragment, illustrating slope and Concordia intercepts of each array. B: Probability density plot of the lower intercepts of each leachate-pair discordia array from (A) with the Concordia curve, plotted along with the nominal dates of known regional thermal events after Spaggiari (2007b) and the approximate depositional age of the Jack Hills metaconglomerate (Spaggiari, 2007a). Arrays defined by L1-L2 leachate pairs have lower intercepts near 0 to 0.5 Ga, while L2-R and (where three leaching steps were conducted) L3-R pairs define arrays with much older – largely Archean – lower intercepts



other minerals: monazite responds poorly to annealing (Peterman et al., 2012), while baddeleyite ($ZrO_2$) displays complicated behavior upon abrasion despite its chemical similarity to zircon (Rioux et al., 2010).

Even more puzzling, modern (zero-age) Pb loss is ubiquitous in zircon (Stern et al., 1966; Black, 1987; Hansen and Friderichsen, 1989; Hansen et al., 1989) and to a lesser degree baddeleyite (Reischmann, 1995; Söderlund et al., 2004; Rioux et al.,
2010), even when it is not observed in other minerals such as monazite (Black, 1987) and sphene (Hansen et al., 1989) from the same sample. Despite some early suggestions, laboratory handling has been largely ruled out as a source of such zero-age Pb-loss (Black, 1987); much to the contrary, laboratory acid treatment reproducibly *decreases* normal discordance both in zircon and other minerals (Mattinson, 2005; Rioux et al., 2010; Peterman et al., 2012). Even in unannealed zircon, where leaching may induce unwanted isotopic fractionation, leachates are consistently more discordant than residues (Mattinson, 1994; Davis
and Krogh, 2001; Mattinson, 2011). Clearly, fully understanding these phenomena is central to the reliability of chemically abraded zircon U-Pb ages.

Consistent with literature expectations (Mattinson, 2005, 2011), chemical abrasion is remarkably successful at removing Jack Hills zircon domains that have undergone open system processes: concordance consistently increases with increasing leaching extent (Figures 2, 3). Components removed in the first four hours (L1) are observed to cluster in an array near the
origin in Figure 2, suggesting they have previously undergone near zero-age Pb loss. Chemically, these components are not stoichiometric zircon, with zirconium representing less than 90% of the cation mass budget. Instead, we suggest that both highly metamict (amorphous) (e.g., Holland and Gottfried, 1955; Utsunomiya et al., 2004) zircon, as well as inclusions and crack-filling precipitates of other less durable minerals, are rapidly dissolved and removed within these first four hours of chemical abrasion. The geochemistry of material removed during subsequent abrasion steps is markedly closer to that of pristine zircon,
though still detectably altered according to the *LREE-I* alteration index of Bell et al. (2016, 2019). Consequently, we suggest that this material removed in the later hours of chemical abrasion corresponds to variably radiation-damaged, altered zircon that may have been been partially annealed or recrystallized.

To better understand the age of open-system behavior affecting discordant leachate fractions, in Figure 6A we estimate the vector of Pb-loss removed by a single leaching step by plotting discordia arrays defined by sequential analysis pairs for the
same fragment. Ordered by leaching step in Concordia space, the lower intercept age of Pb-loss removed by chemical abrasion steadily increases with leaching extent. In particular, two modes are observed: near zero-age lower intercepts corresponding to L1 leaching steps, and broadly Archean lower intercepts corresponding to later leaching steps (Figure 6B). This trend, along with the relatively pristine zircon chemistry of later leaching steps, may be explained by considering that zircon domains which have undergone ancient but *not* recent lead loss must have been subsequently partially annealed or recrystallized. Such domains
would consequently be more resistant to chemical abrasion than their fully metamict counterparts, and thus preferentially accessed only in the later stages of chemical abrasion.

These results, in the context of other recent observations, support the conclusion of Black (1987) that zero-age lead loss in zircon results from aqueous processes associated with exposure and incipient weathering. The Jack Hills zircons have not been affected by any recent tectonothermal disturbances (Spaggiari, 2007a, b), and (according to lithium zonation) have never been
metamorphosed above greenschist facies (Trail et al., 2016) – yet they still display pervasive recent and ancient lead loss. In





contrast to the terrestrial Jack Hills zircons dated here, Lunar zircons of equivalent antiquity display little to no Pb-loss even in leachates (Barboni et al., 2017) despite potential exposure to shock metamorphism (Crow et al., 2018); one of the clearest distinguishing factors to explain this discrepancy is the complete absence of water on the moon.

While diffusion of most cations (including U and Pb) in zircon is extraordinary slow (Cherniak, 2003), disordered, partially
metamict zircon has long been known to be susceptible to aqueous alteration via recrystallization – on laboratory timescales at hydrothermal temperatures (Pidgeon et al., 1966; Geisler et al., 2001, 2003a, b, 2004), and over longer timescales even at ambient temperatures (Stern et al., 1966; Black, 1987; Tromans, 2006; Delattre et al., 2007). There is no reason to expect other accessory minerals such as monazite or sphene to be immune from analogous recrystallization processes (Harlov et al., 2010; Gysi et al., 2018). Instead, we suggest that partial resetting of zircon and baddeleyite during aqueous recrystallization may
simply reflect the extreme incompatibility of Pb in the zircon (and baddeleyite) crystal under natural conditions (Watson et al., 1997). In this context the comparative immunity of higher-$Pb_c$ minerals like monazite and sphene to such exposure-related resetting may simply reflect *closed-system* aqueous recrystallization enabled by their higher tolerance for $Pb^{2+}$ substitution. Exposure-associated aqueous recrystallization may thus satisfactorily explain both the ubiquity of young Pb-loss in terrestrial zircon and baddeleyite, as well as its absence in monazite and sphene.

Why, then, does chemical abrasion succeed for zircon but not baddeleyite? When heated above $\sim$800 °C at atmospheric pressure, metamict zircon decomposes into microcrystalline $ZrO_2$ and $SiO_2$, the latter of which is partially volatilized (Nasdala et al., 2002; Váczi et al., 2009). This process is evidently sensitive to the crystallinity and surface area of the zircon in question, and forms the basis for the whole-grain direct evaporation technique of Kober (1986), which, while limited to unspiked $^{207}Pb/^{206}Pb$ ages, may resolve zircon domains with different Pb-loss histories in a manner reminiscent of chemical
abrasion. Notably, baddeleyite is dramatically more acid-soluble than zircon under laboratory conditions (Rioux et al., 2010). By converting metamict zircon to quantitatively acid-soluble $ZrO_2$ crystallites, low-pressure high-temperature (LPHT) annealing prepares zircons for acid leaching. The high temperature at which this conversion occurs and the subsequent quantitative dissolution of $ZrO_2$ crystallites may explain why isotopic fractionation and reverse discordance are rare in chemical abrasion with annealed zircon but can be significant when acid leaching baddeleyite (Rioux et al., 2010), monazite (Peterman et al.,
2012), or un-annealed zircon (Davis and Krogh, 2001). Since baddeleyite is already $ZrO_2$, there is no way to preferentially increase the solubility of damaged domains by annealing, and thus no way to avoid isotopic fractionation, as observed by Rioux et al. (2010), during incomplete hydrothermal dissolution.

Critically, since low-temperature aqueous recrystallization products appear to remain rather poorly crystalline (featuring microlites, nanopores, and residual amorphous zones (Geisler et al., 2003b, 2004; Delattre et al., 2007; Hay et al., 2009)),
we may expect that partially metamict zircon that has undergone exposure-associated aqueous Pb-loss and recrystallization remains susceptible to oxide decomposition during LPHT annealing. If this interpretation is correct, the absence of isotopic fractionation during chemical abrasion of zircon may depend more upon the decomposition of radiation-damaged zircon during annealing than on the direct acid-solubility of metamict zircon.





## 4.2 Geological History of Hadean Jack Hills Zircons

Despite the limited metamorphic grade of the Jack Hills conglomerate (Spaggiari, 2007a, b; Trail et al., 2016), All zircon fragments we analyzed show clear chemical signs of alteration in leachate fractions, with enrichments in LREE, U, and Th – corresponding to low *LREE-I* in the "altered" field of Bell et al. (2016, 2019). In L1 leachates, which also display relatively low Zr cation proportions, the extreme enrichments in LREE, U, and Th may be attributed in part to inclusions or crack-filling secondary minerals. The more modest enrichments in L2 and L3 leachates are more likely attributable to partially metamict zircon. This latter case leads unavoidably to some ambiguity regarding the origin of the atypical chemistry of these leachates: if certain zones in a given zircon are preferentially metamict, they must have crystallized with particularly high U and Th concentrations. However, since magmatic zircon has not been observed to crystallize with high LREE, we may assume these contaminants were added at or near the time that Pb was lost from the metamict source domains of L2 and L3 leachates. Fortunately, the two independent decay chains of the U-Pb system allow us to estimate the timing of this alteration.

While highly heterogeneous, the lower intercepts of leachate pairs may be crudely divided between two modes: one modern and one Archean (Figure 6). The complete decoupling of the major L1-L2 Pb-loss mode from any known regional metamorphic events in the Narryer terrane is consistent with the hypothesis that this represents aqueous recrystallization during modern exposure and weathering. In this context, it may be significant that the lower intercepts of L2/3 - R pairs broadly scatter around the estimated depositional age of the Jack Hills quartzite, with a mean lower intercept of 3050 Ma.

While the discordia arrays defined by successive leaching steps are subject to substantial interpretive uncertainty (and need not be geologically meaningful considering the possibility of time-transgressive Pb-loss), it is nonetheless apparent from Figure 6 that L2 and L3 domains do not appear to have been heavily influenced by the same zero-age Pb-loss process seen in L1 domains – suggesting that such domains are not as metamict as they once were. Consequently, it appears that either ancient low-grade metamorphic events or prolonged burial may have acted to partially anneal these domains, locking in ancient Pb-loss. In other words, regional metamorphic events in the Narryer terrane appear, if anything, to halt – not initiate – Pb loss. Subaerial exposure and aqueous weathering – not metamorphism – may explain modern and ancient open-system behavior in the Jack Hills zircons. Such a model parsimoniously reconciles the complicated multiple-Pb-loss history of the Jack Hills zircons (e.g., Figure 2) with their relative lack (Trail et al., 2016) of high-grade metamorphism.

Finally, concordant Jack Hills zircon residues that have survived chemical abrasion still display dramatic age heterogeneity, with a 50 Ma range observed between different fragments of the same zircon, as seen in Figure 2C. While chemical abrasion may imperfectly or incompletely remove domains that have undergone ancient open-system behavior, any modern U or Pb loss or addition would occur along a markedly steeper line in $^{206}Pb/^{238}U$ – $^{207}Pb/^{235}U$ space, and thus cannot explain the observed age heterogeneity in RSES58 z6.10. Nonetheless, due to the minimal curvature of Concordia over this age range, we cannot rule out early (>4 Ga) open-system behavior as a cause of this dispersion, even with ID-TIMS precision on the <0.05% level. Considering the infeasibility of high-temperature diffusive daughter loss without dissolution and recrystallization below zircon saturation temperature (Cherniak et al., 1997; Cherniak, 2003; Boehnke et al., 2013; Keller et al., 2017), we are left with two endmember scenarios to explain the observed age heterogeneity in RSES58 z6.10: (1) high temperature overgrowth, and (2)





low temperature recrystallization. The former suggests repeated magmatic or orogenic events within the Hadean; the latter likely requires the presence of liquid water.

## 5 Conclusions

Stepwise CA-ID-TIMS-TEA analyses confirm the Hadean SIMS ages of Jack Hills zircon fragments, while providing insight
into both the geological history of open-system behavior in the Jack Hills zircons and the operation and effectiveness of the zircon chemical abrasion procedure of Mattinson (2005). Jack Hills zircon residues and leachates exhibit complex discordance suggesting at least two recorded modes of post-Hadean Pb-loss, as well as at least one episode of Hadean recrystallization or overgrowth. Concordant Hadean residues reveal 50 Myr of age heterogeneity in the fragments of RSES 58 z6.10, suggesting this single zircon may have experienced multiple episodes of magmatism within the Hadean.

Most Pb-loss in the Hadean Jack Hills zircons studied here substantially post-dates the Hadean, with episodes focused around ∼0 and ∼3 Ga – potentially ameliorating some concerns about the impact of Pb-loss on the Hadean Hf isotope record. Moreover, such Pb-loss does not appear to be driven by high-temperature metamorphism; on the contrary, regional metamorphic events of the Narryer terrane appear to correlate with the partial *annealing* of ancient radiation damage, halting and locking in evidence of ancient Pb-loss in L3 and L2 – but not L1 – domains. Instead, following Stern et al. (1966) and Black (1987), we
propose that Pb-loss in metamict zircon domains is frequently a result of low temperature aqueous recrystallization associated with weathering and subaerial exposure.

While small-scale aqueous recrystallization might well be envisioned as a closed-system process for many minerals, we further propose that the extreme incompatibility of Pb in zircon and baddeleyite ensures that Pb is excluded during aqueous recrystallization. Hence, zero-age Pb-loss is apparent in zircon and baddeleyite even when it is absent in, e.g., coexisting sphene
or monazite. Considering the central role of water in this mechanism of Pb-loss, this hypothesis may explain the ubiquity of recent Pb-loss in terrestrial – but not Lunar – zircon.

Our isotopic and trace element results are consistent with the prior expectation that chemical abrasion (Mattinson, 2005) effectively removes zircon domains that have undergone partial open-system behavior, including both metamict zircon and contaminating inclusions. Over the course of twelve hours of HF leaching, leachate chemistry evolves from U, Th, and LREE-
enriched towards normal zircon, and from discordant to concordant. While the first (L1) leachates are the most radiogenic, they are also the most discordant, and reflect the youngest Pb-loss (Figure 6). The cation proportion of Zr is diminished only in L1 leachates, suggesting most inclusions are removed in the first four hours of chemical abrasion. Meanwhile, elevated U and Th contents in leachates are consistent with the hypothesis that chemical abrasion preferentially removes the same metamict domains that are susceptible to Pb-loss through aqueous recrystallization. We further note that the conversion of partially
metamict zircon to dramatically more acid-soluble baddeleyite during LPHT annealing may explain why it is only for zircon that chemical abrasion is successful in removing open-system domains without significant isotopic fractionation.

Finally, we find that the *LREE-I* alteration index of Bell et al. (2016, 2019) accurately identifies non-primary geochemistry in discordant leachates. In particular, these results demonstrate that the trace element ratio cutoffs defined by Bell et al. (2016)



to identify alteration via SIMS are also applicable to trace element concentrations determined by ICP-MS in the TIMS-TEA (Schoene et al., 2010) workflow. Consequently, we hypothesize that screening *in situ* analyses by *LREE-I* on a cycle-by-cycle basis (with, e.g., split stream techniques) may allow *in situ* U-Pb analyses to reject the same altered domains that are removed by chemical abrasion in CA-TIMS.

5   *Code and data availability.*   All code and data is available at https://github.com/brenhinkeller/JackHillsTIMS-TEA

*Author contributions.*   All authors participated in the design of the experiment and interpretation of the results. CBK., PB, and BS conducted the analyses. CBK generated the figures and prepared the manuscript.

*Competing interests.*   The authors declare no competing interests.

*Acknowledgements.*   Thanks to Kyle M. Samperton for valuable discussion. CBK was supported in part by the U.S. Department of Energy
10   under contract DE-FG02-97ER25308.



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
