# Peer review of "Stepwise chemical abrasion ID-TIMS-TEA of microfractured Hadean zircon"

_Geochronology, 2019_

## Referee Comment (RC1) · Urs Schaltegger (Referee) · 15 Jun 2019

This is a great and long overdue study. We are aware since long that SIMS analyses will never be able to resolve single orogenic events and associated multi-episodic zircon growth/overprints in the Hadean. I suggest publication with minor revision and tried to make a series of comments, coded by page/line number below.

The only generally negative aspect I would put forward is the random fragmentation technique using a tip. Controlled, CL image-based fragmentation into single growth domains would have brought more control on the potential mixing of growth domains and would potentially help to interpret the 50 Ma age dispersion of the concordant residue analyses of sample z6.10. But, yes, you can always go to higher complexity of

[Figure]

the analytical workflow.

Comments indicated by Page/Line number: 2/33: "early Pb-loss": maybe explain this, non-zero? 3/6: Eight orders, this is too long ago, pure hstory. I suggest to stay rather with sound, present-day comparisons. 3/10: 1 $\mu$g fragment, add a normal U concentration range for non-specialists 3/13: I would suggest to cite Widmann et al. 2019, sorry for this...  3/23: refer to Figure 1 3/29: My point, uncontrolled fragmentation 5/20: Mark the 4004.2 Ma point with an arrow on Fig. 2B 8/7: 90% of the observed "total" cation budget? 8/20: Figure 5 is isleading, suggesting that you measured leachates by SIMS on first view. Maybe only add "on residues" on the y-axis title? Lines 20—25 good be better written, is not entirely coherent. What did you measure, what did you plot in addition, etc... 8/30: Here lacks somehow an argument in-between these two sentences: the double 238, 235 decay allows determination of of age of lead loss through the Discordia – why then you want to get rid of lead loss through chemical abrasion? 9/caption figure 4: L1 leachates are mainly enriched in LREE, not MREE. You also don't mention inclusions of REE-enriched minerals. 11/5: There is quite a bit of literature explaining this phenomenon, especially for monazite. Watch out for papers by Seydoux, Poitrasson, Cocherie..... 11/26: Would need some explanation how these modes were calculated any Monte carlo – bootstrapping? I would be curious what intercept clusters you obtain when using the Davies et al. 2018 G3 code. 11/33: Lack of tectonothermal disturbances: this would be after incorporation into the sediment, it is thus not true only for the zircons but also for the sedimentary host rock, right? 12/9-10: Don't fully understand the logic of this sentence (Instead....) 12/14: There have been some other arguments in the literature, based on the zircon bulk modulus (e.g., Hazen & Finger, 1979). They are based on the fact that zircon has $SiO_4$ tetrahedra, monazite not (titanite yes...). The zircon lattice is isodesmic and very stiff. Maybe digging a bit into crystal-physics literature..? 12/18-27: Quite ad-hoc explanation and not really in the scope of this paper. Maybe my suggestion above will lead you to some more sound explanations, badd has no isolated [IV]cation tetrahedra – O bonds. 12/33: The crystal domains that remained below the first percolation point do not get

decomposed through the annealing, I would argue, only the ones above. Arguments you may find in Widmann et al. 2019. 13/20: How metamict were they? They are not clustering at around 3.0 Ga, which would indicate fluid-aided lead loss at that time (erosion-deposition), which they shoud if they were metamict. Not sure about your statement. I agree that they were more damaged than they are now. I very much agree with the general statement of lines 24/25. 14/25: This is entirely consistent with the improvement of the Raman parameters found by Widmann et al. 2019

Well done, congratulations!

––––––––––––––––––––––––––––––––

---

## Referee Comment (RC2) · Anonymous Referee #2 · 20 Jul 2019

This is an interesting, well planned and well executed study. While its contribution to understanding the geological history of Jack Hills detrital zircons, the world's oldest known minerals, is relatively modest, the paper is valuable for exploring the new ways of extracting most information from zircon. I believe the paper can be published after moderate revision.

Comments linked to the text:

p.3 lines 14-15. "frequently presumed" by whom? There are several studies after Mattinson 2005 and Mundil et al. 2004 where the effects and conditions of chemical abrasion are explored in greater detail: Mattinson 2011 Canadian Journal of Earth Sciences, 48, 95–105; Huyskens et al. 2016 Chemical Geology 438, 25–35; Widmann et al. 2019 Chemical Geology 511, 1–10.

[Figure]

p.3 line 16. A more appropriate study to compare to is the paper by Amelin 1998 Chemical Geology 146, 25-38, where multiple fragments of 14 Jack Hills grains were dated by U-Pb ID-TIMS using air abrasion and some HF leaching. That study was indeed done before the advent of chemical abrasion, but it has substantial similarity in concept to this one, and I think it would be wrong to ignore it.

p.4 lines 16-17. Are you using both 3M HCl and 3.1M HCl? I doubt it âŸž)) Please correct the wrong number.

p.5 line 12. You can also consider the second study of 238U/235U in zircon by Livermore et al. 2018 Geochimica et Cosmochimica Acta 237, 171–183.

p.5 line 14. Strictly speaking, the Zr concentration in zircon depends on the Zr/Hf ratio, but this is a small change in normalisation (not really necessary to change).

p.5 lines 14-17. This way of getting Th/U rations does not make much sense to me. You can get these ratios independently from measured concentrations of both elements (by either ICPMS or ID-TIMS), and from Pb-isotopic systematics, and compare the value. This gives useful information about the open system behaviour in the U-Th-Pb system.

p.11 lines 21-22. Please take a look at the paper by Widmann et al. 2019 mentioned above where partially dissolved zircons produced by sequential leaching steps of chemical abrasion were studied by Raman spectrometry. This could help you to refine this speculation.

p.11 lines 27-31. I cannot remember the exact paper(s), but I am sure that ancient natural annealing of zircons during metamorphism and its effect of temporary suppressing the loss of radiogenic Pb, until the radiation damage builds up again, have been discussed before.

p.12 line 3. I doubt that it is correct to speak of "complete absence of water on the moon". "Low abundance of water" is more accurate.

p.12 line 4. Do you mean "in crystalline zircon"?
p212 lines 9-14. About sphene (btw you should use its proper name "titanite") and e.g. apatite I would agree. But monazite? AFAIK its content of Pbc is as low as in zircon and baddeleyite. So I think there should be another explanation.

p.12 line 15 on. The temperature of decomposition of metamict zircon clearly depends on the degree of metamictisation. How metamict is the zircon that decomposes at 800°C?

p. 12 line 20. Baddeleyite is more easily soluble than zircon in HF. Direct extrapolation of this difference to response of these minerals to natural processes is unfounded (at least without additional tests) because the composition of the fluids, temperature and duration of exposure are quite different.

––––––––––––––––––––––––––––––

---

## Author Comment (AC1) · 27 Sep 2019

We appreciate the thorough and positive review by both reviewers, and respond here to any comments from Prof. Schaltegger that required changes to the text. Quotes from the review are italicized.

*This is a great and long overdue study. We are aware since long that SIMS analyses will never be able to resolve single orogenic events and associated multi-episodic zircon growth/overprints in the Hadean. I suggest publication with minor revision and tried to make a series of comments, coded by page/line number below. The only generally negative aspect I would put forward is the random fragmentation technique using a tip. Controlled, CL*

[Figure]

*image-based fragmentation into single growth domains would have brought more control on the potential mixing of growth domains and would potentially help to interpret the 50 Ma age dispersion of the concordant residue analyses of sample z6.10. But, yes, you can always go to higher complexity of the analytical workflow.*

More controlled zircon microsampling would certainly be an excellent avenue for future work. Though one can exert *some* control with a sufficiently fine carbide tip by scoring the zircon where you wish it to break, slicing by laser or FIB would allow much more precise (if more expensive) selection of zircon domains for TIMS analysis.

*Comments indicated by Page/Line number:*
*2/33: "early Pb-loss": maybe explain this, non-zero?*
*3/6: Eight orders, this is too long ago, pure history. I suggest to stay rather with sound, present-day comparisons.*
*3/10: 1 $\mu$g fragment, add a normal U concentration range for non-specialists*
*3/13: I would suggest to cite Widmann et al. 2019, sorry for this...*
*3/23: refer to Figure 1*

We have modified the text accordingly.

*3/29: My point, uncontrolled fragmentation*

Yes, it largely was. Next time!

*5/20: Mark the 4004.2 Ma point with an arrow on Fig. 2B*

We have added rectangles highlighting the concordant residues to make this point more visible.

*8/7: 90% of the observed "total" cation budget?*

We have changed "observed" to "measured" to reflect that this only includes cations whose concentrations were determined in our analyses.

*8/20: Figure 5 is misleading, suggesting that you measured leachates by SIMS on first view. Maybe only add "on residues" on the y-axis title? Lines 20-25 good be better written, is not entirely coherent. What did you measure, what did you plot in addition, etc...*

Good suggestion, thank you. We have attempted to clarify.

*8/30: Here lacks somehow an argument in-between these two sentences: the double 238, 235 decay allows determination of age of lead loss through the Discordia – why then you want to get rid of lead loss through chemical abrasion?*

We have expanded on this in the revised text.

*9/caption figure 4: L1 leachates are mainly enriched in LREE, not MREE. You also don't mention inclusions of REE-enriched minerals.*

We have attempted to clarify: while MREE are much less enriched than L1 leachates, they are still enriched substantially above residues, and Eu is *less* enriched than the adjacent MREE. This apparent preference for, e.g., 3+ Gd over 2+ Eu would not seem to be expected if the host were a purely structureless, metamict glass.

*11/5: There is quite a bit of literature explaining this phenomenon, especially for monazite. Watch out for papers by Seydoux, Poitrasson,*

*Cocherie...*
*11/26: Would need some explanation how these modes were calculated*
*any Monte carlo – bootstrapping? I would be curious what intercept clus-*
*ters you obtain when using the Davies et al. 2018 G3 code.*

We have expanded these sections. For the latter, yes: Monte Carlo resampling.

*11/33: Lack of tectonothermal disturbances: this would be after incorpora-*
*tion into the sediment, it is thus not true only for the zircons but also for the*
*sedimentary host rock, right?*

True!

*12/9-10: Don't fully understand the logic of this sentence (Instead...)*
*12/14: There have been some other arguments in the literature, based on*
*the zircon bulk modulus (e.g., Hazen & Finger, 1979). They are based on*
*the fact that zircon has SiO4 tetrahedra, monazite not (titanite yes...). The*
*zircon lattice is isodesmic and very stiff. Maybe digging a bit into crystal-*
*physics literature..?*
*12/18-27: Quite ad-hoc explanation and not really in the scope of this pa-*
*per. Maybe my suggestion above will lead you to some more sound expla-*
*nations, badd has no isolated [IV]cation tetrahedra – O bonds.*
*12/33: The crystal domains that remained below the first percolation point*
*do not get decomposed through the annealing, I would argue, only the ones*
*above. Arguments you may find in Widmann et al. 2019.*

We have attempted to clarify and reduce the speculativeness of this section.

*13/20: How metamict were they? They are not clustering at around 3.0 Ga,*
*which would indicate fluid-aided lead loss at that time (erosion-deposition),*

*which they should if they were metamict. Not sure about your statement. I agree that they were more damaged than they are now. I very much agree with the general statement of lines 24/25.*

Presumably not ever fully metamict I would agree! Just somewhat damaged..

*14/25: This is entirely consistent with the improvement of the Raman parameters found by Widmann et al. 2019.*

Agreed!

---

## Author Comment (AC2) · 28 Sep 2019

We appreciate the thorough and positive review by both reviewers, and respond here to any comments from Reviewer 2 that required changes to the text. Quotes from the review are italicized.

*This is an interesting, well planned and well executed study. While its contri-bution to understanding the geological history of Jack Hills detrital zircons, the world's oldest known minerals, is relatively modest, the paper is valu-able for exploring the new ways of extracting most information from zircon. I believe the paper can be published after moderate revision.*

[Figure]

*Comments linked to the text:*
*p.3 lines 14-15. "frequently presumed" by whom? There are several studies*
*after Mattinson 2005 and Mundil et al. 2004 where the effects and condi-*
*tions of chemical abrasion are explored in greater detail: Mattinson 2011*
*Canadian Journal of Earth Sciences, 48, 95–105; Huyskens et al. 2016*
*Chemical Geology 438, 25–35; Widmann et al. 2019 Chemical Geology*
*511, 1–10.*

There is indeed more literature here. The "frequently presumed" comment is pointed at the fact that in practice, many of us in the zircon TIMS community routinely chemically abrade for 12 hours (at either 180 or 210) and hope for the best unless obvious signs of Pb-loss are observed in the resulting data; we ourselves are as guilty of this as any.

*p.3 line 16. A more appropriate study to compare to is the paper by Amelin*
*1998 Chemical Geology 146, 25-38, where multiple fragments of 14 Jack*
*Hills grains were dated by U-Pb ID-TIMS using air abrasion and some HF*
*leaching. That study was indeed done before the advent of chemical abra-*
*sion, but it has substantial similarity in concept to this one, and I think it*
*would be wrong to ignore it.*

You're right; we originally cited Amelin 1999 Nature, but we have added a citation of this earlier paper as well.

*p.4 lines 16-17. Are you using both 3M HCl and 3.1M HCl? I doubt it.*
*Please correct the wrong number.*

We have standardized on 3 M; this imprecision in terminology comes from the fact that while 3.0 M HCl is applied by dropper bottle, the columns at this point have been previously conditioned with 6.0 M HCl, so the first eluent is slightly stronger than the

3.0 M HCl added; this is phenomenon is sometimes acknowledged (in lab chemistry manuals, etc.) by referring to the resulting eluent "3.1 M".

> *p.5 line 12. You can also consider the second study of 238U/235U in zircon by Livermore et al. 2018 Geochimica et Cosmochimica Acta 237, 171–183.*

Yes, the Livermore et al. value of $137.817 \pm 0.031$ is quite consistent with the Hiess et al. value of $137.818 \pm 0.045$.

> *p.5 line 14. Strictly speaking, the Zr concentration in zircon depends on the Zr/Hf ratio, but this is a small change in normalisation (not really necessary to change).*
> *p.5 lines 14-17. This way of getting Th/U rations does not make much sense to me. You can get these ratios independently from measured concentrations of both elements (by either ICPMS or ID-TIMS), and from Pb-isotopic systematics, and compare the value. This gives useful information about the open system behaviour in the U-Th-Pb system.*

Yes indeed, though this will be below analytical uncertainty for the Zr/Hf. To avoid this approach for Th/U, one would need (at minimum) a Th isotope dilution spike: all U is consumed by the TIMS analysis, so there is no U in the TEA solutions by which Th/U could be determined by ICPMS.

> *p.11 lines 21-22. Please take a look at the paper by Widmann et al. 2019 mentioned above where partially dissolved zircons produced by sequential leaching steps of chemical abrasion were studied by Raman spectrometry. This could help you to refine this speculation.*

Thanks! This paper came to our attention only after submission, but is indeed quite relevant.

*p.11 lines 27-31. I cannot remember the exact paper(s), but I am sure that ancient natural annealing of zircons during metamorphism and its effect of temporary suppressing the loss of radiogenic Pb, until the radiation damage builds up again, have been discussed before.*

Seems likely, if perhaps not in this exact context.

*p.12 line 3. I doubt that it is correct to speak of "complete absence of water on the moon". "Low abundance of water" is more accurate.*

Fair enough – "near absence"

*p.12 line 4. Do you mean "in crystalline zircon"?*

Yes, we will add this clarification.

*p212 lines 9-14. About sphene (btw you should use its proper name "titanite") and e.g. apatite I would agree. But monazite? AFAIK its content of Pbc is as low as in zircon and baddeleyite. So I think there should be another explanation.*

While monazite certainly has lower Pbc than sphene or apatite, there seems to be some evidence (e.g. Catlos and Miller 2008, doi.org/10.2475/05.2016.03) that monazite Pbc can be quite variable and in any case higher than zircon.
Nonetheless, we have tried to reduce the speculativeness of this section.

*p.12 line 15 on. The temperature of decomposition of metamict zircon clearly depends on the degree of metamictisation. How metamict is the zircon that decomposes at 800 C?*

Fully metamict, generally. However, as Vaczi et al (2009) note, the details can be complicated, especially for partially metamict zircon: "it is far from uncommon to observe that zircon breaks down at temperatures well below the thermodynamically predicted decomposition temperature. There appears to be no well-defined temperature for the onset of decomposition."

*p. 12 line 20. Baddeleyite is more easily soluble than zircon in HF. Direct extrapolation of this difference to response of these minerals to natural processes is unfounded (at least without additional tests) because the composition of the fluids, temperature and duration of exposure are quite different.*

We have reduced the speculation here, but the solubility in HCl (not HF) is more likely applicable to natural solutions; here the difference between zircon and baddeleyite is fairly stark.